# WNT Activation and TGFβ-Smad Inhibition Potentiate Stemness of Mammalian Auditory Neuroprogenitors for High-Throughput Generation of Functional Auditory Neurons In Vitro

**DOI:** 10.3390/cells11152431

**Published:** 2022-08-05

**Authors:** Francis Rousset, Giulia Schilardi, Stéphanie Sgroi, German Nacher-Soler, Rebecca Sipione, Sonja Kleinlogel, Pascal Senn

**Affiliations:** 1The Inner Ear and Olfaction Lab, Department of Pathology and Immunology, Faculty of Medicine, University of Geneva, 1206 Geneva, Switzerland; 2Institute of Physiology, Department of Biomedical Research (DBMR), University of Bern, 3012 Bern, Switzerland; 3Department of Clinical Neurosciences, Service of ORL and Head and Neck Surgery, University Hospital of Geneva, 1205 Geneva, Switzerland

**Keywords:** otic progenitors, auditory neuroprogenitors, auditory neurons, auditory progenitors stemness, auditory neuron regeneration, in vitro model, 3R

## Abstract

Hearing loss affects over 460 million people worldwide and is a major socioeconomic burden. Both genetic and environmental factors (i.e., noise overexposure, ototoxic drug treatment and ageing), promote the irreversible degeneration of cochlear hair cells and associated auditory neurons, leading to sensorineural hearing loss. In contrast to birds, fish and amphibians, the mammalian inner ear is virtually unable to regenerate due to the limited stemness of auditory progenitors, and no causal treatment is able to prevent or reverse hearing loss. As of today, a main limitation for the development of otoprotective or otoregenerative therapies is the lack of efficient preclinical models compatible with high-throughput screening of drug candidates. Currently, the research field mainly relies on primary organotypic inner ear cultures, resulting in high variability, low throughput, high associated costs and ethical concerns. We previously identified and characterized the phoenix auditory neuroprogenitors (ANPGs) as highly proliferative progenitor cells isolated from the A/J mouse cochlea. In the present study, we aim at identifying the signaling pathways responsible for the intrinsic high stemness of phoenix ANPGs. A transcriptomic comparison of traditionally low-stemness ANPGs, isolated from C57Bl/6 and A/J mice at early passages, and high-stemness phoenix ANPGs was performed, allowing the identification of several differentially expressed pathways. Based on differentially regulated pathways, we developed a reprogramming protocol to induce high stemness in presenescent ANPGs (i.e., from C57Bl6 mouse). The pharmacological combination of the WNT agonist (CHIR99021) and TGFβ/Smad inhibitors (LDN193189 and SB431542) resulted in a dramatic increase in presenescent neurosphere growth, and the possibility to expand ANPGs is virtually limitless. As with the phoenix ANPGs, stemness-induced ANPGs could be frozen and thawed, enabling distribution to other laboratories. Importantly, even after 20 passages, stemness-induced ANPGs retained their ability to differentiate into electrophysiologically mature type I auditory neurons. Both stemness-induced and phoenix ANPGs resolve a main bottleneck in the field, allowing efficient, high-throughput, low-cost and 3R-compatible in vitro screening of otoprotective and otoregenerative drug candidates. This study may also add new perspectives to the field of inner ear regeneration.

## 1. Introduction

Sensorineural hearing loss is the most frequent sensory disorder in humans, affecting over 460 million people worldwide or about 5% of the world population [1]. In addition to genetic susceptibilities, many environmental factors are known to generate permanent damage to the sensory cells of the cochlea. Hair cells and spiral ganglion neurons, forming the auditory synapse, are particularly vulnerable and unable to regenerate in mammals. Therefore, the loss of these inner ear cell-types leads to irreversible sensorineural hearing loss. Despite the pandemic scale of the problem, there is no primary cure for this highly disabling condition. Hearing aids and cochlear implants alleviate the symptoms but are not causal therapy and are not suitable for all patients. Any causal therapy for hearing loss would have a tremendous impact for affected individuals and for the society as a whole and would address a true unmet clinical need.

In contrast to humans and mammals, lower vertebrates such as birds, amphibians and fish can efficiently regenerate sensory hair cells in both the auditory and vestibular parts of the inner ear, leading to restoration of hearing and balance [2,3,4,5]. In mammals, limited hair cell regeneration occurs in the vestibular system of rodents [6,7,8,9,10]. However, no spontaneous regeneration has been observed in the mature auditory system [11,12,13]. Regeneration of auditory neurons was also described in rodents and humans as early as 2005 [14].

However, common to all discoveries in the field of auditory neuroregeneration is the fact that mammalian cochlear progenitors retain only a minimal regenerative potential into adulthood and that inner ear progenitors reach senescence after two to five passages in sphere-forming suspension-culture assays [15]. This regenerative potential is mainly observed during the first post-natal week in the mouse, however, rapidly declines with age [16,17]. The reason of this early senescence is still poorly understood, and overcoming this barrier could have major consequences for the development of future regenerative therapies. Another important reason delaying the development of otoprotective and otoregenerative treatments is the lack of robust in vitro models compatible with high-throughput screening. As a consequence, research in the field mainly relies on primary organotypic cultures, resulting in high variability, low throughput and high associated costs and ethical concerns.

Our recent discovery of ANPGs derived from the A/J mouse with an unprecedented capacity to form neurospheres and to proliferate beyond 40 passages [18] offers now new possibilities to address many of the aforementioned limitations in parallel. Phoenix ANPGs are isolated as primary cells from the A/J mouse cochlea at P5-P7, a developmental stage characterized by the spontaneous spiking of auditory neurons [19]. Phoenix ANPGs exhibit nearly unlimited intrinsic self-renewal properties, thus, providing virtually endless material, dramatically reducing the number of animals needed for experimentation [18].

In the present study, we aimed at reprogramming presenescent ANPGs (i.e., from C57Bl/6) in order to prolong their stemness based on active pathways in phoenix cells. For this purpose, we performed a transcriptomic comparison of phoenix ANPGs to low-stemness “classical” ANPGs isolated from the C57Bl/6 mouse and early passage A/J. We hypothesized that differentially expressed genes and pathways would carry important roles in ANPGs self-renewal and stemness. Two main differentially regulated pathways,—namely, the TGFβ smad and WNT pathways—were enriched in low- and high-stemness ANPGs, respectively.

Following pharmacological activation of WNT and inhibition of the TGFβ smad pathway, C57Bl/6 ANPGs exhibited dramatically increased neurosphere growth and passage numbers, whilst expressing main markers of auditory stem cells, including Sox2 and Nestin. Importantly, even at high passage numbers (>20), the so-called stemness-induced ANPGs (si-ANPGs) were still able to differentiate into functional spiral ganglion-like cells expressing both, markers of auditory neurons and supporting glial cells.

Together, the present study demonstrates successful reprogramming of auditory neuroprogenitors to enhance their regeneration potential in vitro. Simultaneous activation of the WNT pathway and dual smad inhibition may be a promising step toward cochlear regeneration and hearing restoration in patients.

## 2. Materials and Methods

### 2.1. Spiral Ganglion Cells Isolation, In Vitro Culture and Differentiation

Collection of mouse inner ear spiral ganglion cells was done as previously described in [18,20,21]. Days 5–7 postnatal A.B6-*Tyr*^+^/J (Stock No: 002565, Jackson; carrying the wild type allele of *Tyr* in a A/J strain genetic background (Stock No: 000646) and C57Bl6/J (Stock No: 000664) pups were used. Tissue dissociation was achieved following enzymatic digestion (StemPro™ Accutase™ Cell Dissociation Reagent) and mechanical pipetting trituration.

Isolated cells were maintained in culture with proliferation media consisting in DMEM:F12 with 15 mM HEPES buffer and 2 mM L-glutamine supplemented with 1× N2 and B27 supplement (Thermo Fisher, Zug, Switzerland), 1× Penicillin streptomycin (100 U/mL) (Thermo Fisher, Zug, Switzerland), in the presence of bFGF (10 ng/mL; ProSpec, Luzern, Switzerland), IGF1 (50 ng/mL; Cell Guidance Systems, Cambridge, UK), Heparan sulfate (50 ng/mL; Sigma Aldrich, Buchs, Switzerland) and EGF (20 ng/mL; Cell Guidance Systems, Cambridge, UK) in ultra-low attachment six-well plates (Corning, Sigma Aldrich, Buchs, Switzerland) and passaged twice a week using enzymatic digestion with Accutase™ followed by mechanical trituration as previously described [18,21].

For differentiation, both phoenix and si-ANPGs were plated on Matrigel 1/100 coating (hESC qualified; Corning, Buchs, Switzerland) for 7 days with a medium change on day 3 or 4. Differentiation medium consists in DMEM:F12 with 15 mM HEPES buffer and 2 mM L-glutamine supplemented with 1× N2 and B27 supplement (Thermo Fisher, Zug, Switzerland), 1× Penicillin streptomycin (100 U/mL) (Thermo Fisher, Zug, Switzerland), in the presence of BDNF (Cell Guidance systems, Cambridge, UK; 10 ng/mL), NT-3 (Prospec, Luzern, Switzerland 50 ng/mL) and LIF (Cell Guidance systems, Cambridge, UK; 10 ng/mL).

### 2.2. Cell Counting

ANPGs neurospheres were dissociated enzymatically and following mechanical pipetting trituration. They were manually counted using a FAST READ 102^®^ (Biosigma, Cona, Italy) hemocytometer, according to the manufacturer’s instructions. Cell number was determined for three independent cultures at every passage, up to passage 5 or until the ANPGs reach senescence.

### 2.3. Automated Analysis of ANPGs Neurosphere Growth

ANPGs isolated from the P5–7 old C57Bl6/J pups modiolus (see paragraph above) were plated at 10,000 cells/well in a 96 Ultra low attachment well plate. Wells containing ANPGs were organized in triplicates and placed 100 µL DMEM:F12, N2, B27 (negative control), +growth factors (IGF, EGF, FGF and HS), +growth factors + WNT agonist (CHIR99021; Axon Medchem, Groningen, The Netherlands) and or TGFβ smad inhibitors (LDN193189 and SB431542, Axon Medchem, Groningen, The Netherlands). Phoenix ANPGs grown in “state-of-the-art” conditions (medium + growth factors) were used as a positive control. Pictures of ANPGs-derived neurospheres were performed twice a week using ImageXpress plate reader over 38 days and size of the neurospheres was monitored using the MetaXpress software (v6.6.3.55, Molecular Devices, Biberach, Germany).

### 2.4. Cell Cycle Analysis by Flow Cytometry (FACS)

The cell cycle of both phoenix and si-ANPGs was studied by DNA staining using propidium iodide followed by FACS analysis according to previously described protocol [18]. FLOWJO (v10.6.2, Basel, Switzerland) software was used to determine the percentage of proliferating cells (engaged in S or G2-M phases of the cell cycle).

### 2.5. RNA Sequencing

Transcriptomic comparison was performed between low-stemness ANPGs organoids, obtained from C57Bl/6 and A/J (at passage 2) and high-stemness phoenix ANPGs obtained from A/J ANPGs at passage 5. RNA extraction was performed using a Qiagen RNA extraction minikit (Qiagen, Hombrechtikon, Switzerland), according to the manufacturer’s protocol. RNA concentration was determined using a Nanodrop spectrometer and RNA integrity was checked with a the 2100 bioanalyzer (Agilent, Basel, Switzerland), following manufacturer’s instruction (RIN > 9). The RNA library was prepared as previously described and sequenced following *Illumina TruSeq* protocol [18]. Analysis was performed as previously described [18].

The average mapping rate to the UCSC Mus musculus mm10 reference was of 91.78%. 14,407 genes were identified following removal of the poorly or non-expressed genes. The *p* values of differentially expressed gene analysis were corrected for multiple testing errors with a 5% FDR (false discovery rate) using the Benjamini–Hochberg (BH) procedure. Main differentially expressed gene ontologies were determined using G:profiler (https://biit.cs.ut.ee/gprofiler/gost (accessed on 2 January 2021)) and gene ontologies network built using EnrichmentMap (version 3.3.1) and AutoAnnotate (version 1.3.3) applications [22] in the cytoscape software environment [23] as previously described [24]. All RNA-sequencing data files were submitted to the ArrayExpress database at EMBL-EBI under the accession number E-MTAB-11869.

### 2.6. Video Time Lapse Microscopy

si-ANPGs or phoenix cells were differentiated for 7 days on Matrigel 1/100 (hESC qualified; Corning, Buchs, Switzerland) coated six-well plates and loaded with FLUO-8 (Interchim, Basel, Switzerland) according to the manufacturer’s protocol. Differentiated cells were then subjected to glutamate stimulation (100 μM). Ca^2+^ evoked fluorescence kinetics was recorded using a Zeiss Axio Observer Z1 with a Definite Focus 2 microscope as previously described [18].

### 2.7. Immunofluorescence and Confocal Microscopy

Phoenix or si-ANPGs were differentiated for 7 days on Matrigel 1/100 (hESC qualified; Corning, Buchs, Switzerland) coated coverslips. For progenitors, undissociated neurospheres were left to attach on Matrigel 1/100 coated coverslips for 4 h in the incubator. Fixation, permeabilization and immunostaining of the samples were performed as previously described [18]. A list of Antibodies used and dilutions is available in Appendix A.

### 2.8. Electrophysiological Characterization of Differentiated SGNs

We followed the procedures published previously [25]. 80,000 cells were plated in differentiation media on laminin-coated (0.1 mg/mL, Sigma) 14 mm coverslips for electrophysiological recordings. Whole-cell somatic patch-clamp recordings were performed at room temperature using borosilicate glass pipettes (Harvard Apparatus GC150F-10) pulled with a Zeitz DMZ-Universal puller with resistances ranging from 5 to 7 MΩ. Cell somata were identified with an upright microscope (Nikon Eclipse E600FN) (40×, NA 0.80) equipped with an infrared GP-CAM3 Altair Astro camera. The pipette solution contained (in mM): 123 K-gluconate, 7 KCl, 1 MgCl_2_, 5 Na_2_-ATP, 10 EGTA, 10 HEPES; pH 7.35 (KOH), 285–290 mOsm.

The bath solutions contained (in mM): 135 NaCl, 5.8 KCl, 0.9 MgCl_2_, 1.3 CaCl_2_, 5.4 D-glucose, 10 HEPES, 0.7 NaH_2_PO_4_ and 2 Na-pyruvate (pH 7.35). Liquid junction potentials were corrected for all experiments. Signals were amplified with an Axopatch 200B Amplifier, low pass filtered at 5 kHz and digitized at 10 kHz with an Axon Digidata 1320B. The data acquisition and analysis were performed using pClamp software (v10.7 Molecular Devices, Biberach, Germany) and Graphpad (prism). Whole-cell current-clamp experiments were performed with 0 pA holding currents and spiking was initiated by current steps from +5 to +65 pA. Potassium and hyperpolarization-activated cyclic nucleotide–gated (HCN) currents were induced by voltage-steps from −80 mV to +60 mV and −130 mV to 0 mV, respectively, with 10 mV increments. Psora-4 (100 nM, DMSO), Lidocaine (10 nM, ddH_2_O) and TTX (5 µM, ddH_2_O) were prepared in stock solution and maintained at −20 °C.

Drugs were dissolved in the extracellular solution at the working concentration on the day of the experiment. We recorded from differentiated phoenix and si-ANs cells from 12–15 days in culture and did not observe any difference in physiological properties at these time points. We therefore focused on recordings at 12 days in culture for the results presented here. Averages are given as the mean ± SEM. *n* = 33 phoenix and 15 si-ANs.

### 2.9. Statistical Analysis

The Figure 5B was analyzed using one-way analysis of variance followed by Turkey’s multiple-comparisons test. The Figures 5C and 6H were analyzed using two-way analysis of variance followed by Sidak’s multiple-comparisons test. For the Figure 7 and Appendix A, significances were determined using non-parametric Student’s *T* test with Mann-Whitney correction. Statistical analysis was performed with GraphPad Prism software (version 9.2.0). For statistical analysis of RNAseq data, please refer to the paragraph RNA sequencing of the method section. Values with *p* < 0.05 were considered as statistically significant. * *p* < 0.05, ** *p* < 0.01, *** *p* < 0.005, and **** *p* < 0.0005.

## 3. Results

### 3.1. Intrinsic Stemness Properties of ANPGs Isolated from C57Bl/6 and A/J Mice

Following the isolation and in vitro culture of ANPGs from the modiolus of day 5 postnatal mouse pups from C57Bl/6 and A/J mice, we observed important discrepancies in the growth and passaging abilities of neurospheres (Figure 1). Following DNA staining, we performed a cell cycle analysis on auditory neurospheres from both genetic backgrounds (Figure 1C). At early passage (P2), ANPGs derived from C57Bl/6 and A/J mice initially exhibited a relatively similar proportion of proliferating cells of 10% and 15%, respectively. However, whereas C57Bl/6 progenitors reached senescence around passage 3, A/J ANPGs increased their proliferation rate along passages to about 35% at passage 5 (Figure 1C) and beyond [18]. This intriguing observation suggests the existence of a high-stemness subpopulation of progenitors, which is enriched with passages in A/J derived organoids—namely, the phoenix ANPGs [18].

### 3.2. Comparative Transcriptomic Analysis (RNA-Sequencing) of High- vs. Low Stemness Neuroprogenitors

In order to investigate the mechanisms of self-propagation in phoenix auditory neuroprogenitors—but also limiting stemness in C57Bl6—we performed a transcriptomic comparison between A/J mice-derived neurospheres (at passage 5) and presenescent C57Bl/6 (at passage 2) and early passages A/J-derived ANPG neurospheres (at passage 2) (Figure 2 and Figure 3). At early passage (passage 2), ANPGs from C57Bl/6 and A/J mice showed a relatively close pattern of gene expression based on the phylogenetic tree (Figure 2A) and multidimensional plot (Figure 2B). Similarly, a relative conserved pattern of gene expressed in both, early passages C57Bl/6 and A/J can be observed on the heatmap, showing a ranking of all genes of the dataset classified from highest to lowest expression in C57Bl/6 (Figure 2C). 

However, the gene expression profile of high proliferating A/J ANPGs was significantly different (Figure 2A–C). For instance, only 47 transcripts were significantly differentially expressed between C57Bl/6 and A/J at passage 2 (Figure 3A,B), whereas this number rose to about 4000 when the comparison was performed between passage 2 C57Bl/6 (Figure 3A,C) and passage 5 A/J (Figure 3A,D). Interestingly, most of the differentially expressed transcript are common between C57Bl/6 and A/J low passage when compared to A/J at passage 5 (Figure 3E).

Together, the data show a closely related gene expression pattern between C57Bl/6 and A/J, suggesting that intrinsic differences linked to the genetic background are minimal. However, at a later passage, A/J ANPGs (phoenix) exhibit a more distinct transcriptomic pattern, even when compared with early passage ANPGs from the same genetic background. Differentially expressed genes could account for the observed stemness of phoenix ANPGs.

### 3.3. Pathway Enrichment Analysis and Differentially Represented Gene Ontologies between High and Low Stemness ANPGs

To further identify genes or pathways involved in phoenix ANPGs stemness, we performed a pathway enrichment analysis using G:profiler [26]. Consistent with the respective number of differentially expressed genes, many pathways were identified with a relatively high level of significance when comparing phoenix ANPGs to A/J or C57Bl/6 ANPGs at passage 2 (Figure 4; Appendix A). However, no significant differences were observed in pathways expressed in both low passage C57Bl/6 and A/J populations (not shown). As expected, numerous genes ontologies referring to DNA metabolism and cell cycle were enriched in phoenix ANPGs (Figure 4A,B). For instance, 85 genes belonging to the cell cycle gene ontology were upregulated in phoenix vs. passage 2 A/J and C57Bl/6 (Appendix A).

Similarly, genes belonging to cell growth, including C-Myc, Bmi1 and KI67 (Appendix A) and telomere extension (Appendix A) were also significantly enriched in phoenix. Consistently, 42 genes from the ribosome gene ontology, together with six genes involved in the oxidative phosphorylation were significantly enriched in phoenix, indicating a higher need for protein synthesis (Appendix A) and metabolic activity (Appendix A), in line with the increased proliferation rate (Figure 1).

Conversely, numerous ontologies, including genes related to neurogenesis, extracellular matrix or signaling pathways were significantly enriched in both low-stemness ANPGs when compared to phoenix cells (Figure 4C). The gene ontology network highlighted an important relative overexpression of the TGFβ smad pathway-related genes in the low propagating ANPGs, suggesting that smad-related signaling could limit stemness of ANPGs (Figure 4D). Other genes significantly enriched in low-stemness ANPGs belong to ontologies referred to as PNS development (Appendix A), gliogenesis (Appendix A), NAPDH oxidases (Appendix A), TGFβ pathway (Appendix A), inhibitors of the WNT pathway (Appendix A) and TYROBP (Appendix A).

As genes/pathways enriched in low-stemness ANPGs may be stemness-limiting factors, they may constitute promising targets to promote self-regeneration of ANPGs. Of particular importance, repressors of the TGFβ smad pathway might be interesting candidates (Figure 4D; Appendix A). It is also noteworthy to mention that some agonists of the WNT pathway, such as the WNT ligands WNT7a and WNT7b or the otic stemness marker Lgr5 are upregulated in phoenix, whereas WNT repressors (e.g., Dkk3, Ndk2, Frzb) are enriched in low-stemness ANPGs (Appendix A).

### 3.4. Reprogramming Stemness of Presenescent ANPGs

In order to reprogram stemness and enhance self-regeneration of presenescent ANPGs, we therefore used the WNT agonist CHIR99021 and dual SMAD inhibitors LDN193189 and SB431542, alone or in combination (Figure 5A). Low-stemness ANPGs from the C57Bl/6 genetic background were plated at 10,000 ANPG/well in a low attachment 96-well plate and neurosphere growth recorded for 38 days (Figure 5B, Appendix A). To date, the “state-of the art” culture medium for otic progenitors consists in DMEM:F12, supplemented with N2/B27 and growth factors (IGF, EGF, FGF and HS) [15,18,20,26]. Indeed, significant growth could be recorded over 38 days, compared to ANPGs proliferation in the absence of growth factors (negative control) (Figure 5B, Appendix A).

However, this growth was far behind phoenix ANPG neurosphere growth performance (used as a positive control). Dual smad inhibition alone did not impact significantly the growth of ANPGs; furthermore, the treatment led to inhomogeneous neurosphere aggregates visible macroscopically (Appendix A). However, WNT stimulation led to significant increase of the ANPG neurosphere growth when compared to ANPGs cultured in “standard conditions” (GF). Interestingly, the combination of dual smad inhibition and WNT stimulation led to dramatic increase in ANPGs neurosphere growth, approaching phoenix ANPGs performances (Figure 5B and Appendix A).

Mammalian ANPGs can typically propagate for a few generations in vitro; however, they rapidly reach senescence, a phenomenon likely also limiting cochlear regeneration [15]. In our hands, C57Bl/6 ANPGs neurospheres could be enzymatically dissociated into secondary and tertiary neurospheres (Figure 5C). However, from the systematic cell counting performed at every passage, cells failed to effectively propagate, and the overall cell numbers remained around the initial D0 value of 200,000 ANPGs, while dropping after passage 2.

The addition of CHIR did not significantly improve ANPG propagation. However, in the presence of both dual smad inhibitors and WNT agonists, a robust cell expansion was observed (Figure 5C). Most of the stemness-induced (si) ANPGs were Sox2 positive (Figure 5E), as observed in non-induced or CHIR + DS treated C57Bl/6 ANPGs (Figure 5D,E) and phoenix cells [18]. Following dual smad inhibition and WNT activation, a strong KI67 expression was detected in neurospheres (Figure 5E). Note that the KI67 expression was mainly observed, but not limited, to the Sox2 progenitors since some Sox2 negative cells were positively stained for KI67 (Figure 5D–E).

Consistent with the observed phenotype, flow cytometry analysis of the cell cycle following DNA staining highlighted a large proportion of cycling ANPGs (>20%) in CHIR + dual smad treated cells, whereas only G0/G1 senescent cells were detected in untreated C57Bl/6 ANPGs (Figure 5F). Importantly, even following freeze and thaw cycles and removal of reprogramming factors after the 20th passage, the propagation rate of stemness-induced (si) ANPGs remained above 20% (Appendix A). Reprogramming therefore elicited sustained mechanisms, allowing for both dramatic enhancement in neurosphere growth and propagation (passage number).

### 3.5. Phenotype of Stemness-Induced ANPGs (si-ANPGs)

At different passages, si-ANPGs were collected to assess their differentiation capacities and the phenotype of differentiated si-auditory neurons (si-ANs) (Figure 6). At passage 1 (Appendix A), 4 (not shown) and 20 (Figure 6B–E), si-ANPGs neurospheres were compared to si-ANs differentiated for 7 days in differentiation medium. Stemness markers such as Sox2 (Figure 6B), Nestin (Figure 6C) and KI67 (Figure 6E) together with the neuronal marker TUJ1 (Figure 6B–E) and the glial cell marker S100 (Figure 6D) were systematically assessed. At any passage assessed, si-ANPG neurospheres were positive for Sox2, Nestin and Ki67; however, only relatively weak and rare TUJ1 immunostaining was observed. S100 was also slightly expressed and mainly detectable at the neurosphere periphery (Figure 6D).

For neural differentiation, si-ANPGs were plated onto a Matrigel coating, in the absence of growth factors and in the presence of BDNF and NT-3 for 7 days, as previously described [21]. Upon differentiation, TUJ1 expression was strongly induced, together with a dramatic change in cellular shape, most of them exhibiting long bipolar axons and dendrites (Figure 6B’). Although at a lower level, Sox2 remained expressed in some differentiated cells with a bipolar shape, also expressing TUJ1 [27]. There was also a significant downregulation of the neural progenitor Nestin (Figure 6C’) and virtual absence of KI67 positive cells (Figure 6E’) in si-ANs. 

In contrast, the expression of S100 dramatically increased, mainly in TUJ1 negative cells (Figure 6D’). At the functional level, si-AN exhibited strong Ca^2+^ mobilization following glutamate stimulation (Figure 6F,G). In fact, both phoenix and si-ANs exhibited sensitive glutamatergic response within the µM range (0.35 vs. 0.45 µM, respectively) (Figure 6H). However, the amplitude of the glutamate-evoked response was significantly stronger in si-ANs (Figure 6H). Together, the data demonstrate an efficient differentiation of si-ANPGs into glutamatergic neuron-like cells.

### 3.6. Electrophysiological Characterization of Stemness-Induced Auditory Neurons (si-ANs) Differentiated from si-ANPGs

To further characterize the glutamatergic neuron-like cells obtained from si-ANPGs differentiation, in particular their similarity to auditory spiral ganglion neurons (SGNs), we performed whole-cell patch-clamp recordings on day 12 of differentiation when the cells showed typical neuronal bipolar morphologies (Figure 7; Appendix A). We performed recordings from 33 phoenix and from 15 si-ANs, which shared prominent morphological characteristics of primary isolated murine SGNs. The first cell type found upon differentiation of both cell lines corresponds in properties to type I SGNs, the most abundant SGN subpopulation (90–95%) innervating inner hair cells [28].

Type I SGN-like cells of both, si-ANs and phoenix origin, had elongated cell bodies (Figure 7B and Appendix A) and shared similar membrane capacitance (phoenix: 6.4 ± 0.4 pF; Si-ANs: 8.8 ± 0.7 pF), similar membrane potential and similar input resistance (Table 1). They are rapidly adapting neurons that typically spike a single action potential with a prominent “sag” current at the offset of the action potential (Figure 7A). The single spike and voltage “sag” is typical for SGN neurons [29].

The “sag” current is, at least in part, mediated by hyperpolarization-activated and nucleotide-gated (HCN) channels abundantly expressed in si-ANs and phoenix type I cells (Figure 7C). Rapid spike adaptation is an important feature of auditory type I SGN neurons that preserves temporal precision during auditory stimulation. Action potential width and amplitude were similar in type I cells generated from si-ANs and phoenix cells (Table 1); however, both cell lines differed significantly in spike latency (Table 1), where phoenix cells were significantly faster (*p* = 0.04, Appendix A).

Within the phoenix line, we identified a second, much rarer (9%) cell type1 that shared characteristics with primary type II SGNs. Indeed, along the cochlea, type II cells are much rarer (5–10% [28]) and innervate outer hair cells. Type II phoenix cells had a spherical cell body (Appendix A), were smaller (4.8 ± 0.2 pF) and had a hyperpolarized membrane potential (−43.3 ± 1.5 mV) compared to type I cells (Table 1). Type II cells also had a significantly lower input resistance (233 ± 22 MΩ, *p* = 0.0018), significantly shorter spike latencies (*p* = 0.004) and significantly smaller action potential amplitudes (*p* = 0.036) compared to type I phoenix cells (Table 1) and, in particular, fired a sustained train of action potentials (Appendix A).

The action potentials in all generated cells were abolished with the voltage-gated sodium channel inhibitor lidocaine (10 nM) (Figure 7A). In phoenix cells, the action potential amplitude was significantly reduced to 33.0 ±13.5% of its initial size, in Si-ANs cells to 14.9 ± 2.7% (Figure 7A). TTX (5 µM), which only affects TTX-sensitive sodium channels, reduced the action potential amplitude in type I phoenix cells to 67.1 ± 14.9% and blocked action potential propagation in type II cells, in line with the reported expression of TTX-sensitive voltage-gated sodium channels (Nav1.1, Nav1.6 and Nav1.7) on primary SGNs [28] (Appendix A).

Both phoenix and si-ANs type I neurons also expressed delayed rectifier potassium channels, which activated at approximately −45 mV and contained a significant portion of currents carried by the Kv1.3 channel, revealed by current inhibition with the specific Kv1.3 antagonist Psora-4 (Si-ANs: 78% inhibition, *p* = 0.035; phoenix: 21% inhibition) (Figure 7D, Appendix A). In contrast, delayed rectifier potassium channels in type II phoenix cells activated at approximately −20 mV (Appendix A). In addition, the amplitude of the potassium current differed in type I phoenix (1479 ± 416 pA at +60 mV) and type I si-ANs (2219 ± 322 pA at +60 mV) neurons as well as in type I phoenix and type II phoenix cells (627 ± 73 pA at +60 mV).

## 4. Discussion

In the present study, we demonstrated that the intrinsic limited stemness of auditory neuroprogenitors can be dramatically extended in vitro upon pharmacological reprogramming. Based on a transcriptomic comparison between “high-stemness” phoenix ANPGs and classical “low-stemness” ANPGs, we identified the key genetic targets to reactivate stemness pathways into presenescent sensory progenitor cells.

By combining the activation of WNT and dual smad inhibition in presenescent ANPGs both growth and passage numbers can be dramatically enhanced, resulting in almost unlimited expansion of stemness-induced ANPGs. Interestingly, even after extensive amplification (>20 passages) and freeze/thaw cycles, si-ANPGs were able to differentiate into electrophysiologically functional auditory neuron-like cells. This reprogramming method not only opens a novel path toward regenerative therapies for hearing loss but also provides a powerful, low-cost, high-throughput and 3R compatible model as an interesting alternative to primary organotypic cultures.

The pandemic scale of hearing loss necessitates the rapid development and implementation of pertinent strategies for otoregeneration. Auditory neurons are particularly relevant targets in the context of cochlear implantation, which remains, thus far, the only available solution for deaf patients. Therefore, spiral ganglion regeneration has recently gained increased attention as a therapeutic strategy in this context [34]. To date, stem cell transplantation [35,36] and neuronal conversion from glial cells [37,38,39] are the two main strategies employed for SGN regeneration.

For instance, glial cells expressing PLP1 and SOX2 can be efficiently converted into new auditory neurons following overexpression of genetic factors, including the transcription factors Ascl1 and NeuroD1 [39], Neurog1 and NeuroD1 [37] and the RNA binding protein lin-28 [38]. Such a conversion of spiral ganglion glial cells to auditory neurons has also been demonstrated following pharmacological reprogramming [40]. However, a direct conversion (i.e., trans-differentiation) is also possible, thereby, raising the problem of an exhaustion of the supporting cell pool. Our strategy is different from previously described reprogramming methods since it is based on the reactivation of auditory progenitor stemness and self-propagation.

Phoenix ANPGs constitute an unprecedented tool to study regeneration in the mammalian auditory system. Based on the transcriptome of phoenix cells, we rationally implemented pharmacological treatments to extend the stemness of presenescent ANPGs. As expected, many differentially expressed genes could be identified between phoenix and presenescent ANPGs.

However, for safe and efficient reprogramming, a direct activation of genes belonging to the cell growth ontology (i.e., putative oncogenes) carries significant risks of cellular transformation. We therefore focused our interest on genes and/or pathways enriched in presenescent ANPGs. Two pathways—namely, WNT and TGFβ smad—were identified as key regulators of ANPGs stemness. In our hands, the combination of a WNT agonist together with dual smad inhibitors led to a powerful growth induction in presenescent ANPG cells and to the possibility of generating a virtually unlimited number of functional auditory neurons-like cells.

Based on recent data, the WNT pathway plays a crucial role in cochlear development and has been extensively investigated in the context of hair cell regeneration from supporting cells [41,42,43]. For instance, in young postnatal mice, significant hair cell regeneration can be achieved through WNT pathway activation and NOTCH inhibition [44]. Furthermore, a clonal expansion of the WNT-related LGR5 expressing otic progenitors was recently demonstrated, thereby, allowing the in vitro generation of hair cells [42].

The WNT agonist CHIR99021, in combination with other pharmacological treatments, has also been shown to promote the conversion of SOX2 expressing glial cells to auditory neurons [40]; however, its potential to promote ANPG expansion has been poorly studied. In our hands, stimulating the WNT pathway was able to enhance ANPGs growth, however, not to a level sufficient to achieve significant passaging and amplification. In addition to the WNT pathway, inhibition of the TGFβ smad pathway was required for successful amplification of ANPGs.

Members of the TGFβ superfamily have been demonstrated to promote the survival of spiral ganglion neurons in vitro, also potentiating neurotrophic effects mediated by neurotrophins [45,46,47]. More generally, TGFβ signaling appears to play a crucial function at several stages of neurogenesis occurring through development [48] or in adults [49] and was shown to repress neural stem-cell proliferation [50]. We assume therefore that the TGFβ pathway plays a similar role in ANPGs, promoting the differentiation into ANs rather than self-renewal.

The reprogramming of ANPGs through WNT activation combined with dual smad inhibition prolonged the stemness of progenitors, which are able to expand virtually indefinitely and generate new functional auditory neurons. Interestingly, a similar pharmacological combination also led to hair cell progenitor expansion in vitro [51]. Whether the presently described reprogramming method will be successful for spiral ganglion regeneration in vivo however remains to be addressed.

Another advantage of the progenitor cell stemness reprogramming relies in its implementation as a novel in vitro platform with high significance to the 3R principle. Stemness-induced ANPGs, such as phoenix ANPGs [18,26,27], are able to recapitulate in vitro the physiology and pathology of spiral ganglion—a key structure for the hearing function and a target for drugs aiming at protecting or rehabilitating hearing (e.g., cochlear implants). In the absence of relevant in vitro models, reprogramming stemness of ANPGs, therefore, constitutes a robust alternative to animal experimentation. Both stemness-induced and phoenix ANPGs combine assets of the cell line and primary culture, retaining the ability to proliferate and differentiate toward auditory neurons and glial cells without genetic transformation.

The reprogramming of ANPGs, which can be frozen and thawed and differentiated at any tested passage into functional auditory neurons, allows access to a virtually unlimited source of primary cells. In contrast, in our research and similarly to what is described in the literature [15], traditional mammalian auditory progenitors could only be propagated for a few passages before reaching senescence, generally resulting in a poor amplification rate (Figure 5D). Such a robust propagation potential has not been previously described for any kind of mammalian auditory progenitors to the best of our knowledge. Both stemness-induced and phoenix ANPGs are compatible with high-throughput assays [21]. They can also be easily distributed to other labs or industries since they can be frozen and thawed multiple times without the loss of stemness of function.

Inner ear organoids, either derived from iPSC or from ES constitute, thus far, the most relevant alternative to animal experimentation [52,53]. In particular, recent advances in directed differentiation of human iPSC allowed the development of heterotypic cochlear organoids recapitulating the sensory hair cells and associated auditory neuron physiology [54,55].

These organoids allow the study of inner ear development and regeneration, ototoxicity [56] and genetic deafness [57,58] in a humanized biological context. However, the experimental throughput is rather low, and the development of such organoids is time consuming and necessitates adequate technical skills. In contrast, the reprogramming of ANPGs constitutes an efficient method to amplify primary auditory progenitors as a robust high-throughput screening platform for toxicity or regenerative studies, genetic or small molecule-based screens or the further development of cochlear implant technologies, among other possible applications.

The main difference between our si-ANPGs-derived SGN-like cells and the primary SGNs were the depolarized membrane potential and a reduced action potential amplitude (Table 1). Furthermore, no type II-like AN could be detected in si-AN (Appendix A). Reasons for not finding type II si-ANs cells may include the lower number of patched cells (*n* = 15), since type II cells make up only about 5–10% of the SGN population. All other characteristics matched excellently with the properties described for primary murine SGNs. The particularly well-preserved electrophysiological properties of si-ANs, even generated following a 30–40 passage expansion of the si-ANPGs, makes it a particularly attractive model to study the pathophysiological and developmental aspects of the cochlea at high throughput and without animal use.

In conclusion, the presently described reprogramming method to reactivate neuroprogenitors self-renewal is a first demonstration of the plastic stemness of ANPGs, thereby, opening a novel path for auditory neuron regeneration in patients suffering from auditory neuropathy. Further in vivo proof of concept will, however, be required to evaluate the long-term therapeutic suitability of such a reprogramming method. In contrast with an eventual therapeutic applicability, our findings are of immediate relevance to the 3R principles, establishing novel robust alternatives to animal experimentation in the field of auditory neuroscience.

## 5. Patents

Rousset F and Senn P (2022). European Patent Application No. 22167241.3.

## Figures and Tables

**Figure 1 cells-11-02431-f001:**
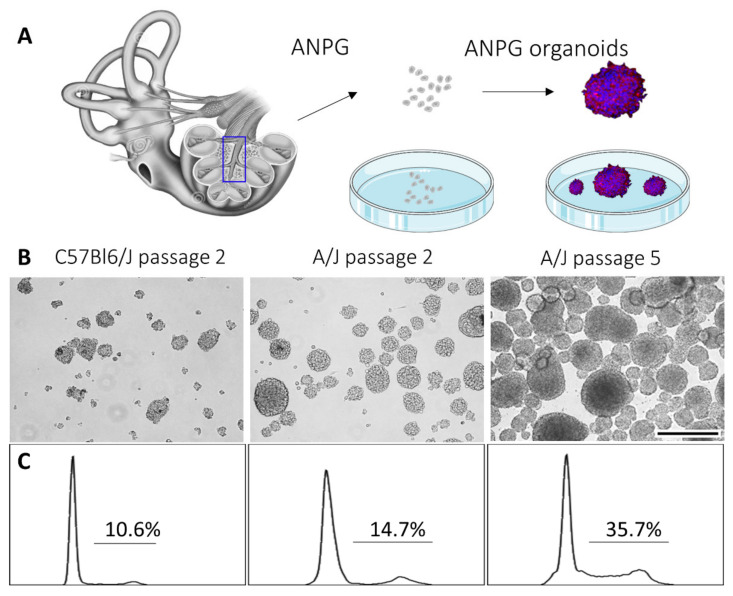
**Generation of ANPGs organoids from C57Bl/6 and A/J mice**. (**A**) ANPGs were isolated from the mouse pup spiral ganglion and cultured as single cell suspensions. Upon growth factor addition (FGF, EGF, IGF and heparan sulfate), ANPGs form organoids. (**B**) Bright-field microscopy pictures of ANPG organoids obtained from C57Bl/6 (left, at passage 2) A/J (middle, at passage 2) and A/J (right, at passage 5). Scale bar 500 µm. (**C**) Cell cycle analysis of ANPG organoids obtained from C57Bl/6 (left, at passage 2) A/J (middle, at passage 2) and A/J (right, at passage 5), following DNA staining. Percentages indicated on the plots represent cells that are actively cycling (in phase S-G2M).

**Figure 2 cells-11-02431-f002:**
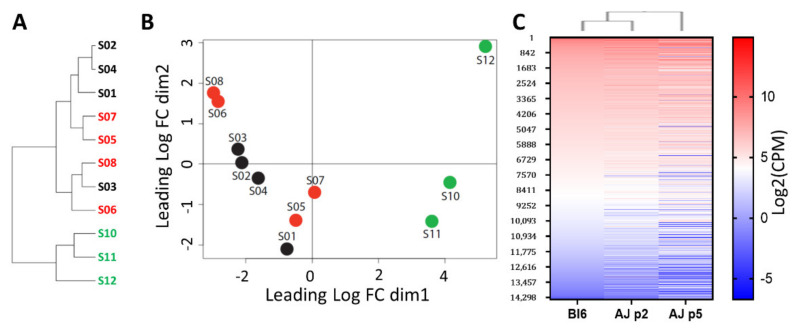
**Relative distance between low and high-stemness ANPG organoids samples**. (**A**) Tree recapitulating relative distance between samples. Low-propagating ANPG: S01 to S04 represent C57Bl/6 passage 2 quadruplicates, S05 to S08 represent A/J passage 2 quadruplicates. High-propagating ANPGs: S10 to S12 represent A/J passage 5 triplicate. (**B**) Multidimensional scanning plot based on the fold changes between samples. Distances between 2 points approximate the expression differences between the corresponding RNA samples. (**C**) Heatmap showing relative gene expression level in low-stemness ANPG from C57Bl/6 passage 2 (lane 1) and A/J passage 2 (lane 2) and high-stemness A/J passage 5 ANPG (lane 3). The data shows relatively similar transcriptomic signature between C57BL/6 and A/J ANPG at passage 2, whereas a more distant pattern of gene expression is observed in A/J at passage 5 (phoenix).

**Figure 3 cells-11-02431-f003:**
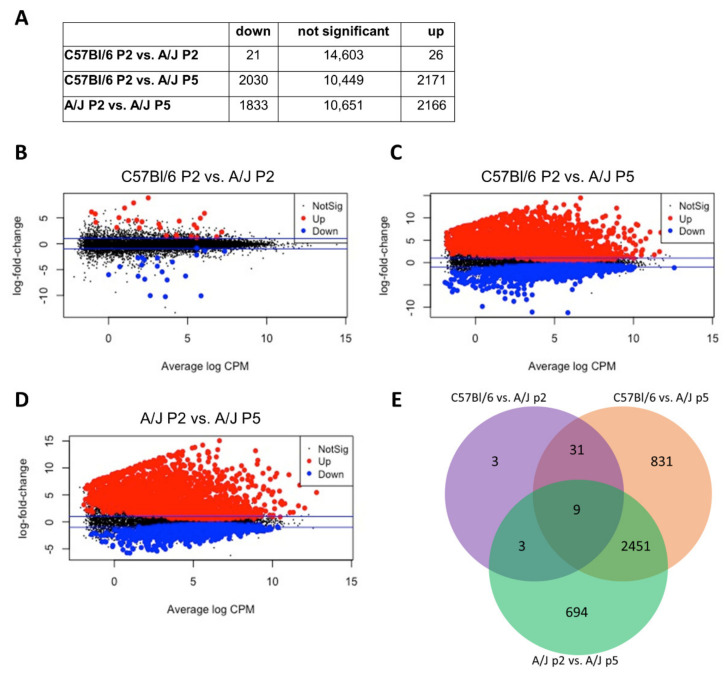
**Bulk RNAseq comparison of low (passage 2 C57Bl/6 and A/J) and high (passage 5 A/J) stemness ANPG**. (**A**) The differentially expressed genes *p*-values are corrected for multiple testing error with a 5% FDR (false discovery rate) following quasi-likelihood statistical test. The correction used is Benjamini–Hochberg (BH). The table gives the differentially expressed genes statistics (FDR 5%) and the number of genes with a fold change >2. (**B**–**D**). Mean difference plots (MD plots) of expression data showing significantly DE genes with a FDR of 5%, highlighted in blue for down and red for up DE genes. The blue line represents the fold change 2 threshold. (**B**) C57Bl/6 P2 vs. A/J P2, (**C**) C57Bl/6 P2 vs. A/J P5 and (**D**) A/J P2 vs. A/J P5. (**E**) Venn diagrams representation of the differentially expressed genes with an FDR < 5% in C57Bl/6 vs. A/J at passage 2 (purple), C57Bl/6 vs. A/J at passage 5 (orange) and A/J at passage 2 vs. A/J at passage 5 (green). Relatively similar gene expression profile are observed between passage 2 A/J and C57Bl/6; however, major changes are observed in passage 5 A/J.

**Figure 4 cells-11-02431-f004:**
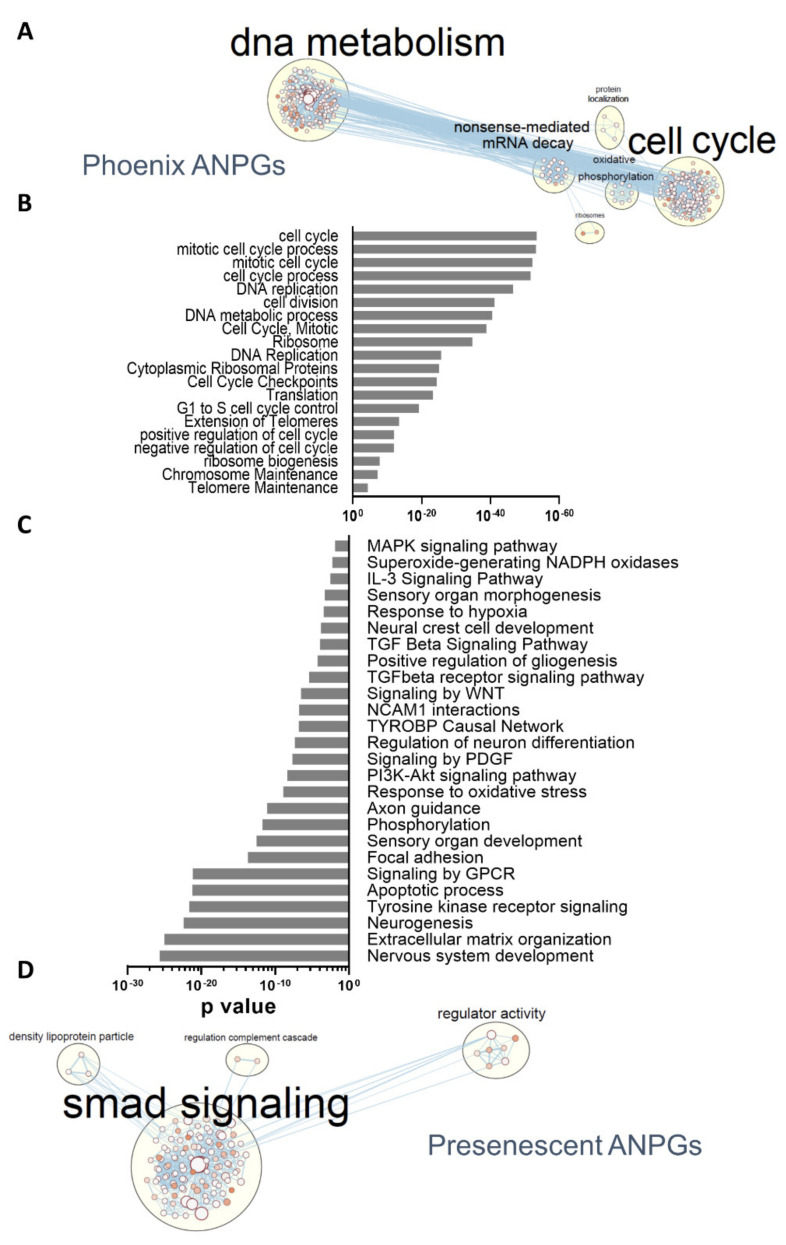
**Differentially represented gene ontologies between phoenix and presenescent ANPGs**. (**A**) Gene ontology network from (**B**). Each node represents a GO term, edges are drawn when there are shared genes between two GO terms. (**B**) Main relevant gene ontologies enriched in phoenix ANPGs vs. C57Bl/6 and A/J ANPGs. (**C**) Main relevant gene ontologies showing a significantly enrichment in C57Bl/6 and A/J ANPGs vs. phoenix ANPGs. (**D**) Gene ontology network from (**C**). Each node represents a GO term, edges are drawn when there are shared genes between two GO terms.

**Figure 5 cells-11-02431-f005:**
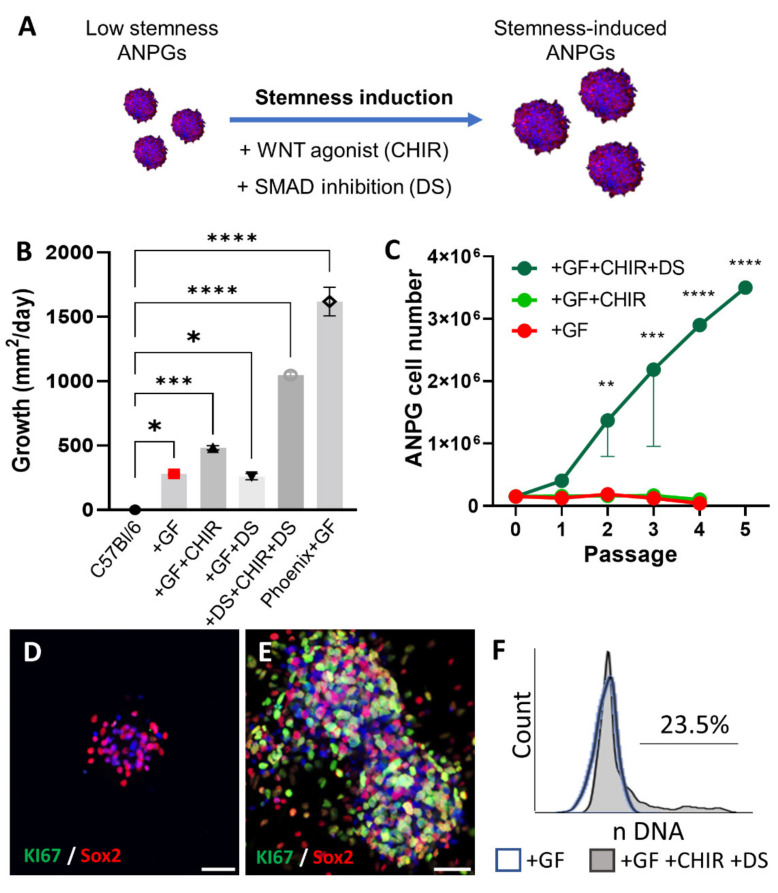
**Neurosphere stemness induction in low propagating ANPGs from a C57Bl/6 mouse**. (**A**) C57Bl/6 ANPG were plated at 10,000/well of a 96-well plate and treated with WNT agonist (CHIR99021; 3 µM) and/or TGFβ Smad antagonist (dual SMAD inhibitors; LDN193189 0.5 µM, SB431542 10 µM) aiming at replicating phoenix ANPG pattern of gene expression and subsequent stemness phenotype. (**B**) Growth of ANPG neurospheres was followed twice a week over 38 days and the bar graph shows the average growth/day. Phoenix cells are used as positive control. C57Bl/6: ANPGs were cultured in DMEM:F12 + N2 and B27 supplements without growth factors. +GF: DMEM:F12 + N2 and B27 + IGF + EFG + HS + FGF (previously described conditions). GF + DS:DMEM:F12 + N2 and B27 + IGF + EFG + HS + FGF + LDN193189 0.5 µM + SB431542 10 µM. GF + CHIR: DMEM:F12 + N2 and B27 + IGF + EFG + HS + FGF + CHIR99021 3 µM. +GF + CHIR + DS: DMEM:F12 + N2 and B27 + IGF + EFG + HS + FGF + LDN193189 0.5 µM + SB431542 10 µM + CHIR99021 3 µM. Phoenix +GF: phoenix ANPG cultured in standard conditions (DMEM:F12 + N2 and B27 + IGF + EFG + HS + FGF). The data represent the average ± SEM of three independent experiments. (**C**) At every passage, following dissociation with Accutase, ANPGs were counted with a fast-read counting chamber. Graph showing the number of cells/passage up to passage 5. Whereas C57Bl/6 ANPGs cultured in standard conditions were not able to expand (see bottom line representing + GF; and middle line representing +GF + CHIR), stemness induced ANPGs exhibited exponential growth (green line (ANPGs + GF + CHIR + DS)). The data represent the average ± SEM of three independent experiments. * *p* < 0.05, ** *p* < 0.01, *** *p* < 0.005, and **** *p* < 0.0005. (**D**,**E**) Immunostaining for the neural otic progenitor marker Sox2 (red) and proliferation marker KI76 (green) in C57Bl/6 ANPG neurospheres cultured in previously described classical conditions (DMEM:F12 + N2 and B27 supplements +IGF +EFG +HS +FGF) (**E**) or reprogrammed with WNT agonist (CHIR99021; 3 µM) and TGFβ Smad antagonist (dual SMAD inhibitors; LDN193189 0.5 µM, SB431542 10 µM). DAPI counterstaining of the nuclei was also performed (blue). Reprogrammed ANPGs exhibit strong KI67 staining demonstrating extensive proliferation. The data are representative from three independent experiments. Scale bars 50 µm. (**F**) Following DNA staining, flow cytometry was performed to determine the percentage of cells engaged in the cell cycle (phase S/G2M; proliferating). Whereas C57Bl/6 ANPGs cultured only in the presence of growth factors (GF) were virtually senescent, ANPGs treated with WNT agonist (CHIR) and dual smad inhibitors (DS) exhibited significant (>20%) fraction of cycling ANPGs. The data are representative from three independent experiments.

**Figure 6 cells-11-02431-f006:**
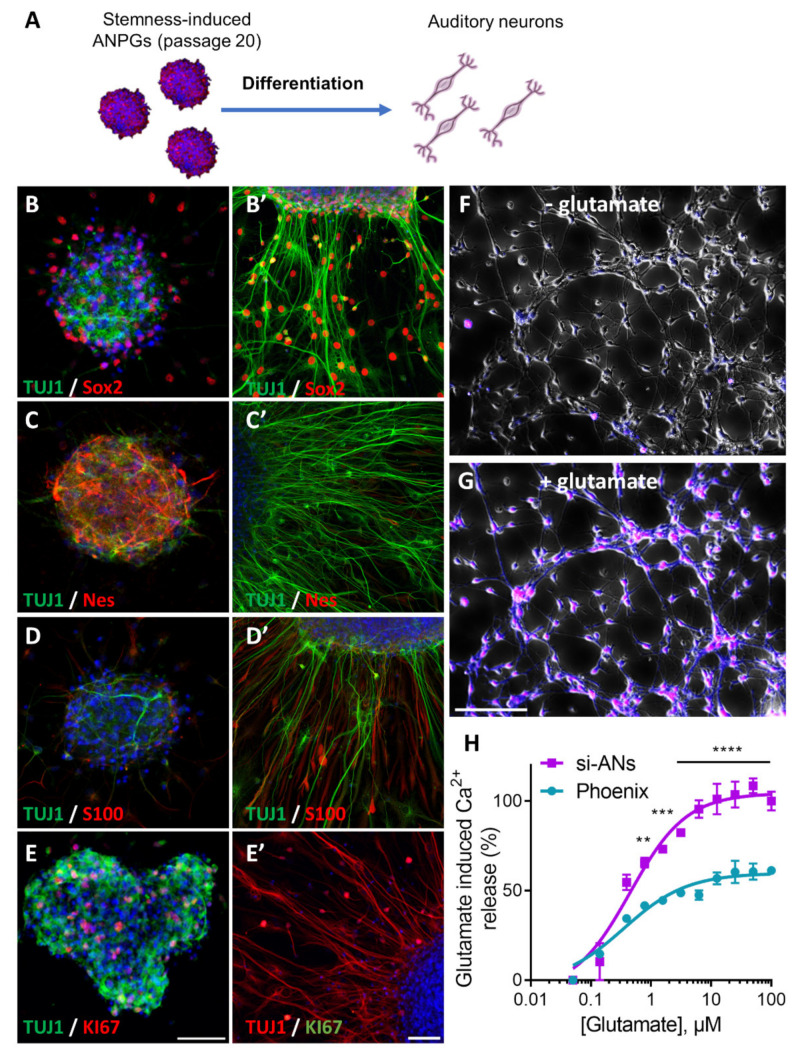
**Phenotypical characterization of stemness induced Auditory Neurons (si-ANs) at passage 20**. (**A**) C57Bl/6 ANPG, were treated with WNT agonist (CHIR99021; 3 µM) or TGFβ smad antagonist (dual SMAD inhibitors; LDN193189 0.5 µM, SB431542 10 µM) aiming at replicating phoenix ANPG stemness phenotype. After reprogramming and an expansion of 20 passages, stemness-induced ANPGs were differentiated on Matrigel coating following removal of mitogenic factors. BDNF (10 ng/mL), NT-3 (50 ng/mL) and LIF (10 ng/mL) were added during the 7 days of differentiation. Immunostainings in differentiated ANs were compared to stemness-induced ANPGs neurospheres at the same passage (**B’**) B-III tubulin (TUJ, green) and Sox2 (red) staining was, respectively, performed in (**B**) ANPGs and (**B’**) ANs. (**C’**) B-III tubulin (TUJ, green) and Nestin (Nes, red) staining was, respectively, performed in (**C**) ANPGs and (**C’**) ANs. (**D’**) B-III tubulin (TUJ, green) and S100 (red) staining was, respectively, performed in (**D**) ANPGs and (**D’**) ANs. (**E’**) B-III tubulin (TUJ) and KI67 staining was, respectively, performed in (**E**) ANPGs and (**E’**) ANs. TUJ staining appears in green in ANPGs and red in ANs and KI67 appears in red in ANPGs and in green in ANs. The results demonstrate strong induction of neuronal markers (TUJ and S100) upon differentiation, whereas the expression of stem cells/neuroprogenitors markers Nestin and the proliferation marker Ki67 strongly decreases. Note that the expression of Sox2 also remains in some differentiating cells, suggesting that a proportion of the stemness-induced ANs are not fully mature. Scale bar (**B**–**E**) and scale bar (**B’**–**E’**) 50 µm. (**F**–**G**) The excitatory potential of si-ANs was tested using live Ca^2+^ imaging. Stemness-induced ANPGs were differentiated for 7 days in absence of growth factors. The resulting ANs Cells were loaded with the Ca^2+^ sensitive ratiometr c probe FLUO-8 and treated with glutamate 100 µM. (**B**) Representative picture of si-ANs before glutamate treatment shows no fluorescence. (**G**) Representative picture of si-ANs 1 s following glutamate addition. Upon glutamate addition, si-ANs exhibit robust Ca^2+^ response represented in purple. (**H**) glutamate-induced Ca^2+^ response in phoenix and si-ANs, following glutamate increments (0–100 µM, 1/2 serial dilutions). Both phoenix and si-ANs exhibit robust Ca^2+^ response with EC_50_ within the µM range. (**B**–**G**) The data are representative from three independent experiments. (**H**) The data represent the average ± SEM of three independent experiments. ** *p* < 0.01, *** *p* < 0.005, and **** *p* < 0.0005.

**Figure 7 cells-11-02431-f007:**
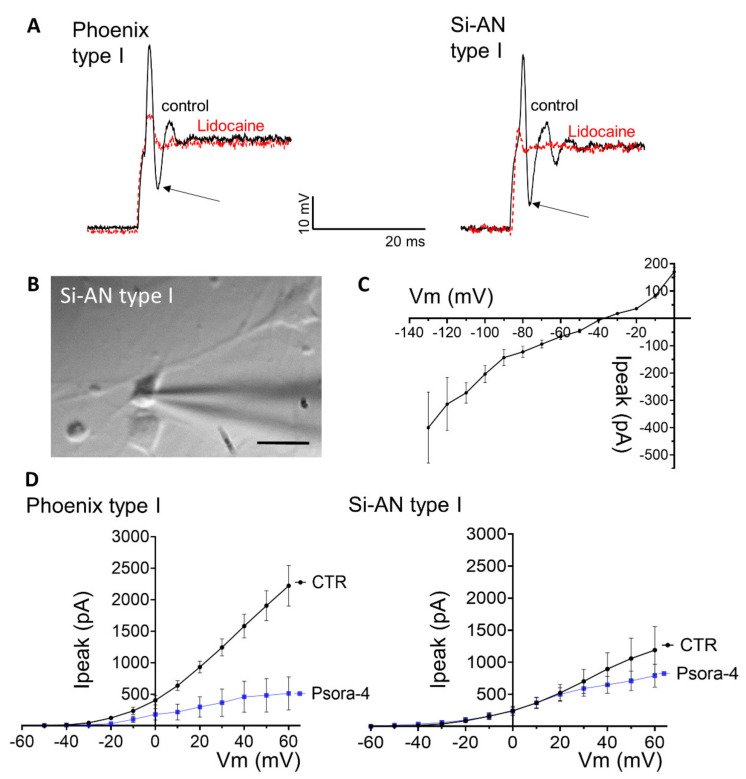
Differentiated phoenix and si-ANs cells functionally and morphologically resemble primary type I spiral ganglion neurons (SGNs). (**A**) Type I phoenix and si-ANs cells show similar rapidly adapting action potentials with a pronounced sag current (arrows) that are effectively blocked by lidocaine (10 nM). (**B**) Morphologically, si-ANs and phoenix type I cells resemble primary type I SGNs. Scale bar 10 µm. (**C**) Voltage–current relationship of HCN channels that are abundantly expressed in type I phoenix and si-ANs cells, leading to a rapid hyperpolarization after the action potential (arrows in A). (**D**) si-ANs and phoenix cells are able to generate large delayed rectifier potassium currents, partially carried by Kv1.3 as block with the specific inhibitor Psora-4 indicates. Psora-4. Shown are voltage–current relationships.

**Table 1 cells-11-02431-t001:** Comparison of electrophysiological properties of phoenix- and Si-AN-derived SGN-like neurons (experimental data) and primary SGNs (values from the literature as indicated).

Cell Type	Phoenix		si-ANs	Primary SGN		Literature 1°SGN
	Type 1	Type 2	Type 1	Type 1	Type 2	
**Abundance**	91%	9%	100%	90–95%	5–10%	[28]
**Vm (mV)**	−40.9 ± 1.5	−43.3 ± 1.5	−40.2 ± 1.9	−60.5 ± 0.9	−65.3 ± 1.7	[30]
**R (input) (M** **Ω)**	438 ± 39	233 ± 22	496 ± 87	apex ~470 300 MΩ 510 ± 70	Basal ~280 200 MΩ 360 ± 120	[31] [28] [32]
**Spike latency (ms)**	3.1 ± 0.4	2.1 ± 0.1	4.4 ± 0.8	4.2 ± 0.8	4.8 ± 0.5	[32]
**AP amplitude (mV)**	34.3 ± 10.7	49.2 ± 11.6	29.0 ± 5.4	50–95		[33]
**AP width (ms)**	2.2 ± 0.2	1.3 ± 0.2	2.5 ± 0.1	Apex: 1.9 ± 0.01 Base: 1.68 ± 0.2	Apex: 1.4 ± 0.02 Base: 1.25 ± 0.03	[28]

*AP*, action potential; *Vm*, resting membrane potential; and *R (input)*, input resistance of the cell.

## Data Availability

The data from the transcriptomic analysis presented in this study are openly available in Array Xpress under the reference number [E-MTAB-11869]. All other data will be made available on request.

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
