# Peer review of "WNT Activation and TGFβ-Smad Inhibition Potentiate Stemness of Mammalian Auditory Neuroprogenitors for High-Throughput Generation of Functional Auditory Neurons In Vitro"

_cells, 2022, doi:10.3390/cells11152431_

Round 1

Reviewer 1 Report

Originality

This study addresses the question of the enhancement of regeneration potential of mammalian inner ear cells as the potential strategy for future regenerative therapy of sensorineural deafness. The aim was to identify signaling pathways responsible for the intrinsic high stemness of phoenix auditory progenitors, isolated from C57B1/6 and A/J mice. A reprogramming protocol for inducing the high stemness in C57B1/6 mice, based on pharmacological combination of the WNT agonist (CHIR99021) and TGFbeta/Smad inhibitors (LDN193189 and SB431542) resulted in dramatic increase  in presenescent neurosphere growth and the possibility to expand auditory neuroprogenitors limitless in vitro, with retained ability to differentiate into electrophysiologically mature type I auditory neurons.

The study may add new prospectives to the field of inner ear regeneration.

- Structure

The article is clearly laid out, with all the key elements present: abstract, introduction, methods, results, discussion, patents, supplementary materials, author contributions, funding, institutional review board statement, data availability statement, acknowledgements, conflict of interest statement, references, table (Table 1), figures (Figure 1-7). Title is straight -to the point and clearly describes the article. 

- Abstract: Abstract is concise and informative. 

- Introduction: Introduction provides the relevant information about the research rationale.

- Methods provide relevant information about the animal models, auditory neuroprogenitor cells  and the methodology of the research.  Data acquisition and analysis are clearly presented.

- Results are clearly laid out and exposed in a logical sequence. Exposition of the results is very well written and keeps the reader involved from the beginning to the end of the article. 

- Discussion: The claims in the Discussion are based on the results. The authors  discuss different forms of auditory cells reprogramming, pointing on the originality of their method based on reactivation of auditory progenitor stemness and self-propagation. Prudently avoiding putative oncogenes, the authors identified  two molecular patways – WNT and TGFbeta smad- as key regulators of auditory neuroprogenitor stemness. In addition to WNT pathway stimulation, inhibition of the TGFbeta-smad pathway was required for successful amplification of auditory neuroprogenitor cells. Characteristics of the newly obtained cells are very promising for future auditory neuroprogenitor cells and spiral ganglion cells studies in vitro. Future in vivo studies are necessary for the evaluation of the therapeutic applicability of these findings. The pandemic scale of hearing loss and secondary pathologies like peripheral tinnitus necessitate the rapid development and implementation of strategies for otoregeneration.

- Ethical Issues: Plagiarism or fraud of any kind could not be noted.

The manuscript is satisfactory in its present form. 

Author Response

We would like to thank the reviewer for the careful review and positive evaluation of our manuscript.

Reviewer 2 Report

My major comment is

Phoenix ANPGs grow great than other strain ANPGs in Fig. 5B even adding WNT activation and Smad inhibition. (Typo in the group DS+CHIR+DS?) It mean phoenix ANPGs harvested from A/J mouse can constitute an unprecedented tool to study regeneration in the mammalian auditory system. I can't understand that you want to show the benefit of wnt and TGF signaling or the Phoenix ANPGs. Why the stemness got better after more passage in Phoenix ANPGs? What is the result of Phoenix ANPGs with GF+CHIR+DS ? I am interested whether the mouse strain has better hearing in phenotype. 

Reviewer 3 Report

The authors studied the possibilities of WNT activation and TGF inhibition in potentiating stemness of mammalian auditory progenitors. It was a novel and interesting study. The findings is important for the future direction of the treatment of sensorineural hearing loss. The methodology has been described in detail. The results were well presented. The conclusion was well supported by the data obtained.

Author Response

We would like to thank the reviewer for the careful evaluation of our manuscript and enthusiasm regarding eventual publication of our manuscript

Reviewer 4 Report

The authors of this study have performed a transcriptomic comparison of auditory neuroprogenitors from C57Bl/6 and A/J mice at early passages with low stemness and auditory neuroprogenitors from A/J mice at later passages with high stemness.

They have identified several differentially expressed genes in 2 pathways and found 3 drugs to regulate WNT and TGFB/Smad pathways to maintain stemness in auditory neuroprogenitors.

Introduction is concise and directly describes the rationale and the hypothesis.

Methods used include:

Spiral ganglion cells isolation and in vitro culture.

Cell counting.

Automated analysis of ANPGs neurosphere growth.

Cell cycle analysis by flow cytometry (FACS).

RNA sequencing

Video time lapse microscopy.

Immunofluorescence and confocal microscopy

Electrophysiological Characterization of differentiated SGNs

This study is brilliant, and the description of the methods facilitate a replication. I have no concerns of the methods or results.

The study  have demonstrated that the stemness of auditory neuroprogenitors can be extended in vitro upon pharmacological reprogramming

Discussion could include a short translational section with current status of human auditory progenitors from iPSC 

Question:

Have the authors tested these drugs in human neuron progrenitors derived from iPSCs?

Reviewer 5 Report

The manuscript " WNT activation and TGFβ-smad inhibition potentiate stemness of mammalian auditory neuroprogenitors for high throughput generation of functional auditory neurons in vitro" explores a robust in vitro orogenic stem cell model allowing high-throughput screening for otoprotective compounds and regeneration. By comparing transcriptomic sequencing, the authors reprogram the currently existing ANPG to extend its stemness based on the activation pathway of Phoenix cells. Activation of WNT and inhibition of the TGFβ/Smad pathway leads to a dramatic increase in prebiotic neurosphere growth and allows for almost unlimited potential for expansion of ANPGs. The reprogrammed cells can also differentiate into neural cells with certain functions, which is expected to provide a new research cell system for the field of inner ear regeneration. This paper has a lot of potential benefits, though it also has several areas of concern. Here are a few major concerns:

1. Please explain whether further transcriptome level or protein level validation was performed after RNA sequencing? The sequencing results showed significant differences in the expression of genes related to gliogenesis and NADPH oxidase in addition to the WNT and TGFβ/Smad pathways; whether validation experiments have been performed on these pathways?

2. In Table 1, when comparing the electrophysiological properties of phoenix and Si-ANs derived SGN-like neurons with Primary SGN, the data of the latter are obtained from the literature, please explain the reasons for not measuring SGN with the same device to eliminate systematic errors?

3. Is there any phenotypic difference between the phoenix ANPGs and the reprogrammed stemness-induced ANPGs?

4. Besides ANPGs, can intervention of WNT and TGFβ/Smad pathways increase the stem proliferation potential of other inner ear progenitors?

5. Please complete the study on the application of stemness-induced ANPGs? For example in a series of throughput screening experiments for highly neurogenic compounds. Are there any commonalities or differences in reprogrammed stemness-induced ANPGs compared to the gold standard spiral ganglion organotypic explant (SGE) model and phoenix ANPGs?

Round 2

Reviewer 2 Report

It is ok.

Reviewer 5 Report

I recommend accepting the manuscript.